# Narrative Review of the COVID-19 Pandemic’s First Two Years in Italy

**DOI:** 10.3390/ijerph192315443

**Published:** 2022-11-22

**Authors:** Flavia Beccia, Andrea Di Pilla, Francesco Andrea Causio, Bruno Federico, Maria Lucia Specchia, Carlo Favaretti, Stefania Boccia, Gianfranco Damiani

**Affiliations:** 1Section of Hygiene, University Department of Life Sciences and Public Health, Università Cattolica del Sacro Cuore, 00168 Rome, Italy; 2Department of Human Sciences, Society and Health, Università degli Studi di Cassino e del Lazio Meridionale, 03043 Cassino, Italy; 3Department of Woman and Child Health and Public Health, Fondazione Policlinico Universitario A. Gemelli IRCCS, 00168 Rome, Italy; 4Centre on Leadership in Medicine, Università Cattolica del Sacro Cuore, 00168 Rome, Italy

**Keywords:** COVID-19, pandemics, Italy, vaccines, distancing, policy

## Abstract

Italy was the first country in the western world to be affected by the COVID-19 pandemic, arguably among the worst-affected ones, counting 12 million cases and 150 thousand deaths two years since the first case. Facing new challenges, Italy has enacted different strategies and policies to limit the spread of the SARS-CoV-2 virus and treat those affected by COVID-19. This narrative review provided an overview of factors, measures, and actions that shaped Italy’s first two years of the COVID-19 pandemic by investigating epidemiological data and using a mixed-method approach. This narrative review aimed to summarize the most relevant aspects and measures and analyze available data to provide policymakers and healthcare providers with the instruments to learn from this pandemic and improve their preparedness for future pandemic events. The first two years of the pandemic differ in that, during the first year, significant necessary changes to the way health systems were organized were implemented, increasing healthcare spending and adopting social and physical distancing measures that were stricter than the ones adopted in the second year. However, as the pandemic progressed, increased knowledge of the virus and related variants, as well as the introduction of highly effective vaccines, which were not equally available to the whole population, resulted in a stratification of COVID-19 infections and deaths based on factors such as age, vaccination status, and individual susceptibility to the virus.

## 1. Introduction

On 31 December 2019, the Wuhan Municipal Health Commission (China) reported a cluster of pneumonia cases of unknown etiology to the World Health Organization (WHO) [1]. A new coronavirus, phylogenetically related to SARS and MERS viruses, was discovered: SARS-CoV-2 [2]. The novel disease caused by the virus was named COVID-19. The rapid spread of cases led the WHO to declare a pandemic in March 2020 [3].

Two years into the pandemic, millions of cases worldwide were registered, and, more specifically, as of 28 February 2022, the WHO had reported 444,872,425 cases and 6,019,792 deaths. Europe is among the worst-affected areas, with 183,218,623 cases and 1,901,657 deaths [https://covid19.who.int/ (accessed on 28 February 2022)]. Italy was one of the first European countries to detect and register cases, and on 28 February 2022, it accounted for 12,782,836 cases and 154,767 deaths. The regions with the highest incidence were Veneto, Friuli-Venezia Giulia, Emilia-Romagna, and Bolzano (Autonomous Province); those with the most increased mortality were Val d’Aosta, Friuli-Venezia Giulia, Lombardia, and Emilia-Romagna [4].

The national government and the regions have put in place measures to prevent intra-national spread, such as lockdowns, reduction of the ability to travel between regions, hygiene measures and social distancing, and trans-national control measures such as airport closures and quarantine for travelers from abroad. In this sense, contact tracing, testing, and quarantine measures were vital, despite the difficulty of discriminating between deaths caused by COVID-19 or simply having it as a comorbidity [5]. The deployed efforts, both at national and regional levels, tried to comply with the Public Health goals during an epidemic: [1] delay the peak and flatten the epidemic curve [2], reduce the overall number of cases [3], and quickly increase the hospital beds on offer (including Intensive Care Unit beds) [6,7].

Alongside the abovementioned policies, population measures were promoted, such as economic aid to the most disadvantaged population groups, families, and businesses, as well as strengthening hospital and territorial services dedicated to the care of COVID-19 patients and the recruitment of dedicated medical and nursing staff.

Then, the advent of vaccination and vaccination programs characterized the second year, with regional plans and the prioritization of administration by age group and at-risk categories, given the shortage of doses and organizational needs, favoring, to a certain extent, the stratification of the pandemic across the country [8].

To understand the extent of the COVID-19 pandemic’s impact on the Italian territory and the most relevant aspects that specifically characterized the pandemic waves in the country, we conducted a narrative review. Several articles were published on pandemic trends and COVID-related issues, each highlighting different features of the Italian response to the pandemic. Articles include information on organizational approaches, epidemiological measurements, superspreading events, and analyses of policymaking preparedness and response, providing valuable contributions but lacking a unified systematization of contents. In this article, we included the evidence we gathered in the scientific literature in a single, harmonized piece of literature, providing an overview of the pandemic in Italy, considering the different factors and the national responses over the first two years. To cope with the dynamic change of pandemic wave definitions over time and different time frames in the literature, the narrative synthesis was enriched with an ex-novo elaboration of surveillance data, adopting a mixed-methods approach. This paper aimed to describe the characteristics of the pandemic in Italy narratively, with the aid of some epidemiological indicators preceding and following the advent of vaccination. Portraying the pandemic in Italy could lead to a better understanding of the dynamics and the challenges the healthcare and society faced, providing policymakers with valuable tools for the future.

## 2. Materials and Methods

This study aimed to provide a concise yet comprehensive overview of factors, measures, and actions that shaped the first two years of the COVID-19 pandemic in Italy. To this purpose, we performed a narrative review to elaborate the events shaping the COVID-19 pandemic in Italy, not necessarily aiming to achieve completeness. In addition, we investigated epidemiological data and analyzed raw data to clarify the work. We searched PubMed to identify articles published in English or Italian from February 2020 to January 2022 that addressed the COVID-19 pandemic in Italy, focusing on the history of the pandemic in the first two years on the national territory and the impact of vaccination. We did not search other databases; because of the number of retrieved articles on PubMed and because we did not aim to achieve completeness in our review, we decided that searching in more databases would have been time-consuming while not adding significant detail to the articles we had already found, which we deemed to be satisfactory for this article.

The following search query was used to evaluate articles on the management of the pandemic and measures implemented in Italy: (“COVID-19” OR “SARS-CoV-2” OR coronavirus) AND (Italy) AND (diffusion OR frequency OR impact OR effect) AND (prevalence OR incidence OR hospitalizations OR mortality OR morbidity OR lethality OR burden OR indicator*) AND (“containment” OR “public health measures” OR vaccin*).

The inclusion criteria were as follows: articles should cover the COVID-19 pandemic response in Italy or describe SARS-CoV-2 and COVID-19 at least partially, be written in English or Italian, and deal with the pandemic as late as January 2022. Any articles not satisfying at least one of these criteria were excluded. Eligibility was evaluated by FB, ADP, and FAC separately. A confrontation between authors solved any mismatches in eligibility evaluation.

The full texts of the articles considered eligible were evaluated.

FB, ADP, and FAC were involved in the analysis of the included articles. Each author analyzed one-third of all the included articles. All three authors extracted relevant information in an Excel document and later integrated it into the final text. The text was then divided into seven sections, which are reported in the Results section of this manuscript as an agreement between FB, ADP, and FAC, as this was decided as the best way to present the results chronologically and logically. The chosen aspects and measures were deemed the most relevant according to the authors’ expertise on the COVID-19 pandemic and enacted countermeasures.

In addition, the bibliography of included articles was screened to retrieve additional relevant publications.

To identify relevant reports and publications, we screened institutional repositories of the Italian Civil Protection Department [9], Italian Ministry of Health [10], Italian Superior Institute of Health (Istituto Superiore di Sanità, ISS) [10], and European Centre for Disease Prevention and Control [11].

To provide a comprehensive overview of epidemiological data and overcome the fragmentation and differences present in literature articles, ADP collected all the available data in the national surveillance system from the ISS website and reports and analyzed them. All the epidemiological data presented in the results derived from the original analysis of national data.

Data about the epidemic and vaccination were obtained from the Civil Protection Department, the Italian Superior Institute of Health, Ministry of Health websites, COVID-19 dashboards, and locally related published studies. These datasets form part of the country’s official public health trace-track surveillance and reporting system on COVID-19 [12]. Figure 1 was created with raw ISS data using Microsoft Excel: data were imported from the xlsx -file “COVID-19 ISS open data” [10] and made into a graph.

Excess mortality data were obtained from the ISTAT website [13], where excess mortality represents the increased percentage of mortality per month compared with the monthly average mortality in 2016–2019. Figure 2 was created on raw ISTAT data using Microsoft Excel, importing data from a database and creating a graph. Data were not further summarized or analyzed [13].

Figure 3 was created with raw ISS data using Microsoft Excel. Data were extracted from ISS Comunicati Stampa n. 14/2021, 20/2021, 24/2021, 30/2021, 35/2021, 37/2021, 42/2021, 69/2021, 71/2021, 03/2022, 08/2022, and 12/2022 [13] into an Excel database, reporting the prevalence of SARS-CoV-2 VOCs in each ISS report, and then a graph was created. Data were not further summarized or analyzed.

The indicators considered in this work were prevalence, crude mortality rate, and apparent crude lethality, in line with data availability and agreement with key epidemic-related indicators reported in literature and reports. The choice was based on the indicators used by scientific literature and institutional organizations to describe COVID-19 epidemiology [14,15,16]. Regarding the vaccination campaign, indicators related to delivered doses to the population were examined.

## 3. Results

When framing the evolution of the first two years of the COVID-19 pandemic in Italy, we found several measures enacted at the local and national levels. We here chose to present the results chronologically, divided into seven sections. The first section, “*Framing of epidemic waves*”, provides an overview of the subsequent pandemic waves that have shaped the pandemic. In contrast, the six sections approach the significant events we identified narratively.

### 3.1. Framing of Epidemic Waves

There is no unambiguous strict definition of an epidemic wave or a specific threshold of cases that marks the beginning of a new wave or just a phase of the previous one in recurrences [17]. Even if operational definitions have been proposed, pandemic waves should not be simplified as incidence curves [18] since other indicators are needed to interpret and create prediction models, considering under-ascertainment and indicators’ limits [19]. Common characteristics are, however, recognizable. A wave implies a pattern of an increasing number of sick individuals, a definite peak, and then a decline. This pattern is typical of some infectious diseases, offering some models of how a condition such as COVID-19 might unfold over time [20].

In the first two years of the epidemic in Italy, four waves were identified, as shown in Figure 1.

The first wave was from February to June 2020, the second one (with two peaks) from October 2020 to July 2021, the third from July 2021 to September 2021, and the fourth from October 2021 to February 2022. 

The second wave was characterized by the highest share of deaths, while most cases were registered during the fourth wave, probably thanks to the spread of more contagious virus variants [21,22].

### 3.2. Towards the Lockdown

On 31 January 2020, despite the initial uncertainty over the definition and classification of cases, the Italian prime minister confirmed that two Chinese tourists were the first two cases of contagion found in Italy, and a state of national emergency was declared [23,24].

The first autochthonous case was reported on 21 February in Codogno, a small town near Milan (Northern Italy), in a 38-year-old man with fever and non-productive cough and no apparent epidemiological link to COVID-19 [25,26]. On the same day, the first death attributable to COVID-19 forced ten towns into lockdown [27].

As positivity notifications multiplied (approximately 1500 new cases within one week, between 24 February and 1 March 2020, and more than 5300 new cases in the next one), several studies were conducted to moderate the uncertainty about when and how the virus started its national and global circulation [24,28,29]. According to some mathematical modeling, it is possible that the diffusion began earlier than in the officially confirmed cases [30,31,32,33]. It has been suggested that the Atalanta vs. Valencia football game, played in Milan on 19 February and having over forty thousand people attending, contributed to the initial spread of the infection on the national territory [34,35]. The direct flights connecting Italy to China, as many Chinese immigrants work in the fashion industry in northern Italy, were also implicated in the early virus introduction and propagation [34]. In addition, the initial COVID-19 case definition that included mandatory epidemiological criteria of history of travel to China resulted in scarce testing and identification, especially of asymptomatic cases that were not likely to seek medical assistance [30].

National databases were created, building surveillance systems through the Civil Protection and the ISS.

Microbiological and epidemiological surveillance of COVID-19 was set up in January, collecting and analyzing information on all SARS-CoV-2 infections confirmed by molecular diagnosis in regional reference laboratories in Italy [36]. On 27 February 2020, epidemiological and microbiological surveillance for COVID-19 was entrusted to the ISS (Integrated Surveillance) [37].

Building on the data available from the Civil Protection since 24 February and the rapid rise in cases up to 9 March 2020 (+3900% in period prevalence), the Italian government declared the extension of the lockdown state at a national level, restricting activities and transfers of citizens, while other countries banned access to Italian travelers, even in the absence of local containment measures or lockdown policies [38,39,40,41].

After continuous growth for nearly four weeks, the reported daily number of COVID-19 cases in Italy peaked from 18–20 April, marking the beginning of the downward phase of the curve.

Along with the lockdown, local and national administrations implemented containment measures to face the epidemic progression across the various Italian regions. These measures combined social distancing, epidemiological and microbiological surveillance and strengthening health services at the national level, recruitment of additional healthcare staff, supplementing the equipment and consumables, and reorganizing territorial and hospital healthcare [42].

As a result of the deployed measures, the first wave’s descending slope bottomed out in May [43].

### 3.3. Organizational Changes in the First Pandemic Year

Three different organizational models built on pre-existing organizations shaped the delivery of care in the Italian regions as a consequence of the Italian National Healthcare System governance: because regions are entitled to manage the healthcare system in their jurisdiction, this resulted in different approaches to common problems caused by the pandemic, depending on the territorial organization of Primary Care and Community Health Services, mainly being hospital-based (e.g., Lombardia, Lazio, Liguria), community-based (e.g., Veneto), and integrated (e.g., Emilia-Romagna) [44].

Regions with advanced structures and experience in primary care, community health services, and home care (traditionally associated with treating and managing chronic diseases) deployed them to address the epidemic, performing better in controlling it and managing public health-related problems [16,45]. These departments were involved in contact-tracing programs, actively calling or sending emails to any retrieved contacts of COVID-19 cases and informing them of behavioral measures to adopt following this notification. 

At least at the initial stage, the epidemic had low spread to the southern regions, probably because of the effectiveness of early containment measures since the first cases were in the northern regions. In addition, broader testing (i.e., a more significant number of swabs tested per case detected) was associated with better epidemic control [46,47]. It is also possible that in areas with a higher frequency of infected individuals, the testing capacity of the health system was overwhelmed [48].

At the national level, the increasing implementation of digital health and the positive examples set by other countries [49] led the national government to implement a national contact tracing app for COVID-19 named Immuni [50]. Despite the expectations, as of 31 December 2021, this app only contributed to identifying 44,880 COVID-19 cases, corresponding to less than 1% of total COVID-19 cases reported in Italy in the same period (5,886,411), with 143,956 notifications being generated overall [51].

### 3.4. The End of the First Year of the Pandemic

During the summer of 2020, the number of cases dropped. The laboratory capacity was increased to perform more tests. In addition, residency grants were increased to cope with the shortage of healthcare personnel [52,53]. Hospitals reorganized the dedicated pathways for COVID-19 patients, and telemedicine was implemented [54,55,56,57].

From August 2020, the number of cases began to rise again, starting the second wave in early October. Regions most affected by the second wave, from October 2020 to July 2021, were Bolzano, Friuli-Venezia-Giulia, Val d’Aosta, Veneto, Emilia-Romagna, Piemonte, and Lombardia. 

During the second wave, patients were younger, with fewer concomitant chronic conditions and a milder clinical presentation. Short-term clinical outcomes were better, with lower mortality rates and more transfers to a general ward [58].

As soon as cases started to rise, policies changed accordingly. Measures implemented at the national level consisted of further restrictions on public and private gatherings, the use of facemasks, and actions for international travelers, including advice to avoid unnecessary travel and to quarantine upon return. 

In August 2020, wearing facemasks was mandatory again between 6 pm and 6 am and outdoors, where social distancing was impossible [59]. Then, starting on 24 September some Italian regions introduced stricter measures, i.e., the mandatory use of facemasks at all times, even outdoors [60,61]. COVID-19 testing was implemented: testing for returning from foreign countries, for patients entering healthcare facilities, for contact with an established or suspected case, and periodic screening for healthcare workers was introduced [62]. Subsequently, the new containment measures carried out by the Decree of the Council of Ministers DPCM on 18 October 2020, and those provided by the Decree released by the Lombardia Region on 16 October 2020, were considered by health experts as too weak to address a second wave effectively [63,64]. In October, the percentage change in cases was 567%, while the ratio of positive cases-to-swabs performed rose from 2.16% to 14.7% [46]. Italy was divided into three risk zones (yellow, orange, and red) of increased restrictions [65]. This approach was updated weekly, and regions were allocated to one of the three tiers based on 21 epidemiological indicators in Table A1, to be found in Appendix A [66]. New restrictions on work, school, travel, trade, and culture were reinforced [65,67].

The surge of cases was probably related to two main factors: the crowded public transport system needed to ensure the operation of schools and businesses and the reopening of companies and restarting ordinary activities [52,68].

While the public expected further restrictive measures and a new lockdown, conflicts and a lack of cooperation between the national government and regions delayed the implementation of containment measures, potentially increasing the burden of distress and mental health outcomes among the public [52], already linked as a reaction to a prolonged state of crisis defined by the WHO with the term “pandemic fatigue” [69].

The first epidemic year (24 February 2020–23 February 2021) accounted for 96,348 deaths, 2,832,162 cases, and 38,533,461 swabs. The excess mortality data could not entirely describe it because they could be flawed by several factors, such as data availability, death for other causes exacerbated by the pandemic, population density, and preparedness of healthcare services [70]. However, according to ISTAT data, there was a percentage increase in mortality of 14.3% in 2020 compared with the 2015–2019 average (Figure 2). 

### 3.5. The Arrival of Vaccines and How They Shaped the Second Pandemic Year

While the first pandemic year was characterized by two waves and the adoption of non-pharmacological interventions to address the pandemic, the second year was marked by vaccination.

Several factors enabled the rapid pace of development of vaccines against COVID-19: prior knowledge of the role of the spike protein in coronavirus pathogenesis and evidence that neutralizing antibody against the spike protein is essential for immunity; the evolution of nucleic acid vaccine technology platforms that allow the creation of vaccines and prompt manufacture of thousands of doses once a genetic sequence is known; and development activities that can be conducted in parallel, rather than sequentially, without increasing risks for study participants [71]. With about 2 million cases and 72 thousand deaths, the vaccination campaign started on 27 December 2020, when Italy received 9750 doses of the Pfizer–BioNTech COVID-19 vaccine [72]. The Italian vaccination campaign was coordinated and managed by the European Union. Moderna and AstraZeneca vaccines were approved in the following two months and started to be administered [73].

During Christmas and New Year festivities, the government issued ten days of national lockdown and four days of partial closing [74]. Criticality was placed on the end-of-year festive season, traditionally associated with activities such as social gatherings, shopping, and traveling, which posed significant additional risks for intensified transmission of SARS-CoV-2. 

The ‘pandemic fatigue’ stressed the situation, with some people becoming demotivated to follow recommended protective measures, especially during this period [75]. During the first year, Italy reported the emergence of pandemic fatigue in the population and other European countries. It is a reaction to the prolonged nature of this crisis and the associated inconvenience and hardship. It probably influenced the spread of the virus and posed a threat to efforts to control it [76]. WHO stated that governments should acknowledge their responsibility to address the factors that lead to fatigue [75].

The vaccination campaign was conducted to prioritize risk groups, starting with health workers, staff, and nursing home residents and then favoring categorization by age group [77]. On 2 January 2021, the “National Strategic Vaccine Plan for the Prevention of SARS-CoV-2 Infections” was adopted by Ministerial Decree. The document summarized the guidelines for ensuring vaccination according to uniform standards. Vaccines were and still are offered for free to the entire population, according to an order of priority, which considers the risk of disease, the types of vaccine, and their availability. The plan also foresaw that an adaptive strategy might have been implemented during the epidemic: recommendations on target groups to be offered vaccination could be subject to change and could be updated according to evolving knowledge and available information [78].

The groups identified for priority vaccination in the early stages were health and social care workers, residents, and staff of residential care facilities for the elderly and older persons. Other population categories to be vaccinated were subsequently scheduled on the increased available vaccine doses.

The underlying objective of this strategy was to help maintain the resilience of health services and increase coverage for people with clinical risk factors as the prevalence of comorbidities increases with age.

Subsequently, as new vaccines were approved, with the Decree of the Minister of Health on 12 March 2021, five risk categories were identified based on frailty and age. School and university staff, Armed Forces, Police and Public Rescue, prison services, and other residential communities were also prioritized [79].

In the months following the start of the vaccination campaign, several studies and reports were published to address the efficacy and safety of the authorized vaccines [80,81]. These reports are relevant for addressing efficacy and safety issues related to the newly approved COVID-19 vaccines, which are highly innovative both because of the technology they are based on (especially the mRNA-based Moderna and Pfizer-BioNTech vaccines) and because of the approval process which brought them to be administered just a few months after the outbreak of the pandemic. 

From December 2020 to 1 March 2022, 136 million doses were administered, with 85.49% of the population receiving at least one dose, 80.2% completing the initial vaccination protocol, and 63.5% receiving a booster dose [81]. The reporting rate of post-vaccination adverse events is 99 per 100,000 doses administered, with 82% of this reporting as non-serious [81]. Despite the arrival of vaccines and the restriction measures, Italy registered a third wave from July to September 2021. 

There was an increase in the incidence of COVID-19 cases and deaths in the younger age groups (0–18 years), and the burden on the prevention departments led to a delay in notification and in updating epidemiological data [82]. Therefore, new measures were implemented, reinforcing lockdown in “red zones” and limiting population movements while promoting COVID-19 testing and face mask policies [83,84]. Mandatory vaccination for healthcare workers was established, specifying the specific cases of medical liability in COVID-19 management [85,86].

To reduce the burden on the healthcare systems and tackle vaccine hesitancy, Italy adopted the COVID-19 Green Certificate or “green pass” in June 2021. In July 2021, responding to the drop in administered vaccine doses and the concomitant increase in cases, new restrictions were added [87], including travel, public transportation, and access to workplaces, schools, universities, health facilities, restaurants, and hotels. It also allowed using certain services and participation in numerous cultural, recreational, and sports activities [88,89]. Over time, different types of green passes have been introduced:-Basic Green pass: certificate of vaccination against COVID-19, negative antigenic or molecular test, or previous recovery from the disease.-Enhanced green pass: certificate attesting vaccination or recovery from previous COVID-19 disease. This certificate was introduced later to increase vaccination acceptance among skeptics by limiting access to specific services and activities only to those showing proof of vaccination, hence not including those with a negative swab, either antigenic or molecular.

As of 1 March 2022, 245,795,605 COVID-19 Green Certifications had been issued. This tool effectively contained the increase in cases during the fourth wave while pushing further segments of the population to get vaccinated [90].

The fourth wave emerged at the end of the summer, with a progressive increase in cases. The government adopted local containment measures to avoid exacerbating the socio-economic crisis while also setting a price cap for the price of swabs and diagnostic tests [85,86,87,88].

The reopening of schools (OM. n.256, 6 August 2021) and return to work were the main drivers of increasing in COVID-19 cases. Concerns arose about granting access to education and employment while protecting the frailest and presenting guidelines and recommendations [91,92,93,94].

Co-circulation of SARS-CoV-2 with influenza and other respiratory viruses [93,94,95], as well as the concerning emergency of COVID-19 variants, led to recommendations for the third dose (booster) and the start of the flu vaccination campaign.

### 3.6. The Emergence and Spread of SARS-CoV-2 Variants

The molecular and genomic characteristics of SARS-CoV-2 have led to several variants, each endowed with peculiar virulence. Since the end of 2020, the identification of SARS-CoV-2 Variants of Concern (VOCs) backed with higher infectivity and pathogenicity than the “original” variant has led to public health measures aimed at an estimation of their presence on the national territory and the adoption of appropriate countermeasures [96].

On 27 January 2021, the launch of the Italian Network for SARS-CoV-2 genotyping and phenotyping and for monitoring the immune response to vaccination was announced, promoted by the Ministry of Health, and coordinated by the ISS to perform surveillance of mutations responsible for emerging infections and evaluation of the efficacy and duration of the protection given by vaccination. 

On 31 January 2021, the Ministry of Health issued new rules for the activity of research and management of contacts of COVID-19 cases, prioritizing the research and management of contacts of COVID-19 variant infection cases, indicating that a retrospective search of contacts should be carried out (over 48 h and up to 14 days before the onset of symptoms or the execution of the swab in the asymptomatic case) and perform a molecular test as soon as possible after their identification and at the fourteenth day of the quarantine, rather than at day 10 (as for infections not related to a SARS-CoV-2 variant). Additionally, the ISS suggested that molecular tests should be “multi-target”, i.e., detect multiple virus genes and not only the spike gene (S) that could provide negative results in a variant. When performing genomic surveillance, sample selection for effective genomic surveillance should be representative of the population (geographic origin and age distribution), with priority given to cases suspected of a variant with high transmissibility or higher severity [97].

The emergence of COVID-19 variants of concern (VOCs) propelled vaccination campaigns trying to prevent and control new waves [98]. The ISS coordinated periodic surveys aimed at mapping the spread of SARS-CoV-2 variants in Italy, precisely: lineage B.1.1.7, “English” or Alpha variant (WHO nomenclature); lineage B.1.351, “South African” or Beta variant (WHO nomenclature); lineage P.1, “Brazilian” or Gamma variant (WHO nomenclature); lineage B.1.617.2, “Indian” or Delta variant (WHO nomenclature); lineage B.1.1.529 or Omicron variant (WHO nomenclature). These investigations involved a sample analysis of RT-PCR-confirmed SARS-CoV-2 virus infection cases, with genomic sequencing and subsequent elaboration of a prevalence estimate of the various variants. The results of the surveys are summarized in Figure 3.

**Figure 3 ijerph-19-15443-f003:**
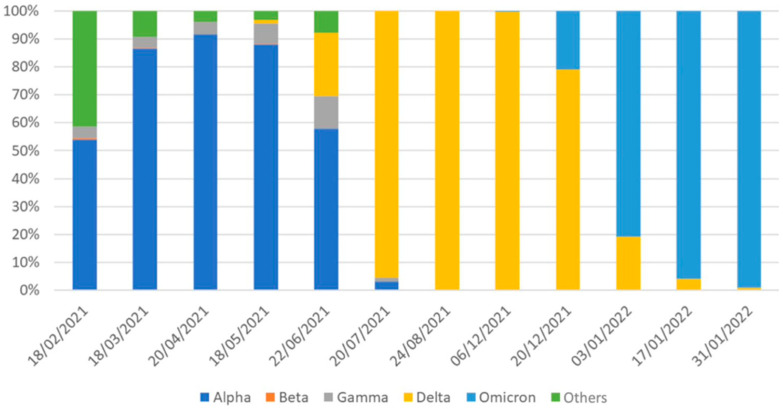
Prevalence of SARS-CoV-2 VOCs. This figure was created elaborating raw data from ISS [99].

### 3.7. Towards “Multiple Pandemics”

After the first pandemic wave, the implications of a decentralized healthcare system and a highly bureaucratic political system with low coordination and conflicts between government, regions, and local authorities resulted in ineffective economic investments and the inability to restructure the Italian healthcare system at territorial and hospital levels. The fragmentation of health policies resulted in different approaches for COVID-19 surveillance (in terms of testing) and hospital management, with many regions initiating the construction of field hospitals and issuing calls for personnel. Although regional and municipal management of state funds successfully provided resources for the most vulnerable segments of the population, different policies across the country often exacerbated social and economic hardship. Regional policy often ran counter to state guidance, leading to confusion and hindering adherence to regulations. The situation seemed to be resolved thanks to the government change in February 2021, when the pandemic management was increasingly centralized. As expressed by Bontempi et al., when a range of possible measures are proposed in distinct regions of the same country, the efficacy of the steps and the credibility of the public administrators regarding health crisis management may be damaged [68].

Parallelly, the stratified access to vaccination and the progressive arrival of new vaccines have led to the fragmentation of the pandemic across the national territory. Additionally, the impact of acute infection on different age groups may have helped stratify the pandemic across the population in Italy.

The consequences of the SARS-CoV-2 infection depend on the subject’s condition, primarily age: an older age exposes one to more serious consequences. Considering the reports published by the ISS throughout the epidemic in Italy, it is clear how age and vaccination status influence infection status and outcomes. Considering the unvaccinated population, the risk of contracting the infection in subjects over 60 years of age was not higher than in younger people, especially given adequate distancing and social protection measures, but the risk of hospitalization and death was significantly higher (relative risk: range 2.4–20, calculated from ISS raw data) [10]. The population receiving a single dose of vaccine (considering a vaccination cycle of at least two doses) presents an overlapping risk. 

In the second year of the pandemic, if we consider a standard population vaccinated with two doses, the risk of hospitalization and death of the population over 60 years of age is still increased, but in a considerably lower way (relative risk: range 1.4–7, calculated from ISS raw data) [10], underlining how the vaccine has acted effectively in risk reduction in the most susceptible populations. 

It is also possible to suggest a temporal pattern in the age composition of the cases diagnosed in Italy: concerning the second wave, in October 2020, about 17% of the new cases were under the age of 18 (school age), while about 11% were over the age of 70. During November and December 2020, these values are reversed, with new cases diagnosed in the population under age 18 falling to around 10% of the total number of new cases, and new cases diagnosed in the population over age 70 rising to 18% of the total number of new cases. With the start of the vaccination campaign, which was organized by identifying the target populations by clinical condition and age, these values are again reversed, a sign of the protective effect vaccines have had on elderly people since the beginning of 2021 [9].

These findings help to hypothesize a model of different, interdependent COVID-19 epidemics for different populations, stratified by age (and vaccination status), to be further investigated in further studies with ad hoc analyses.

## 4. Discussion

The results of our work showed how Italy enacted a series of local and nationwide measures to counter the risks of the COVID-19 pandemic. The COVID-19 pandemic has gradually affected countries worldwide, and Italy has had the ungrateful task of being a pioneer in facing this pandemic. Before mimicking other countries’ approaches (e.g., the lockdown put in place in China), Italy was unprepared in the phase that preceded the national lockdown, as shown in the results of our work, when small countryside towns were isolated while simultaneously allowing large gatherings of people in the nearby city of Bergamo. In addition, the organizational changes in the first pandemic year have shown how having a highly heterogeneous healthcare system can complicate issuing a response to a global shock, as in a pandemic. The pandemic hitting regions in the north first has often been discussed as a protective factor for regions in the south, as explained before. The story of this pandemic is undoubtedly related to the timely distancing measures set in place when SARS-CoV-2 was first identified. Though these were not timely enough to prevent it from spreading in the north, it is agreed that the south was worse equipped and organized to face the storm that hit the north in early 2020. Meanwhile, the national lockdown limited the spread of the virus nationwide, so a proper response could be organized for future waves. In the future, policymakers should consider that the healthcare system fragmentation, which results from Italy’s peculiar history and socio-cultural approach to healthcare, cannot be regarded as adequate without a set of measures to make the system more resilient against global shocks.

The adopted measures showed high variability between the first year of the pandemic when social distancing measures were the primary resources available to prevent people from getting sick, and the second year, when the advent of vaccines made it possible to reshape these measures, incentivizing vaccinations to access services and places that had previously been restricted for sake of public health. When the first year of the pandemic was characterized by pandemic fatigue, this phenomenon was described less often thanks to the lifting of such measures, which was made possible by significant reductions in mortality, despite incidence spiking as a natural consequence. Such differences can be understood as a consequence of the incapability to foresee a pandemic by such a deadly, infectious, and mutagenic virus: the decisions taken by policymakers required a deep understanding of the virus, and the capacity to put common interest before individual freedoms (with freedom of movement being the most limited one) and economic prosperity, taking difficult choices for the sake of public health, to fight a battle that barely anyone could have been prepared for. In the future, policymakers should have an evidence-based approach to complex problems, such as the ones stemming from a new deadly virus rapidly spreading among the population, and make difficult decisions that prioritize people’s health, even when these decisions are unpopular. The difficult choices have paid off: the 2020 lockdown, the social distancing measures, and the Digital Green Certificate have all shown to be effective. Other instruments, such as the app Immuni, were greeted with great hopes but have performed well under expectations. Although promising, digital applications are not always practical as predicted because of relevant limitations related to their nature and end users. For Immuni, the app was conceptually simple to develop, but it was technically unsatisfying. The Digital Green Certificate, which was issued to all countries in the European Union, was more complex from a technical point of view. Still, its effectiveness and ease of use made it a highly adopted and successful instrument. It should be noted that a shaming misinformation campaign welcomed both tools in their early days, which is likely to be repeated for any governmental instrument that is adopted in the future [100,101]. Digital solutions can, hence, be very useful if enacted policies strongly back them, but they should not be regarded as a panacea for all problems and should be accompanied by a critical evaluation of their adoption.

Three key aspects should be considered in preparing for future pandemic waves and in critically reading what has worked and what factors to continue working on: the health, the social, and the economic impact of the pandemic. The loss of one million jobs has been estimated, and emphasis has been placed on the loss of ordinary healthcare services and the effects of quarantine and lockdown on the mental health of children, adolescents, and the elderly.

The strengthening of health systems and the centralization of critical issues, such as health promotion and prevention, must be a priority to avoid a recurrence of an emergency of this magnitude in the coming years. Moreover, tackling misinformation and properly addressing inequalities could reduce vaccine hesitancy and limit the spread of negationism.

Looking at the future, it will be crucial to anticipate pandemics before it is too late, requiring global efforts and collaboration between countries to prevent microbes with pandemic potential from causing a pandemic, necessarily including private partners and stakeholders who have proven to be essential to maximize efforts and achieve challenging goals promptly, as shown in the development process of the COVID-19 vaccines. Citizen information and engagement have often lacked in this pandemic, leading to mistrust and antiscientific behaviors: ensuring a reliable, comprehensive source of news and information for the public will help fight inaccurate news and unify societal behaviors, making it easier to adopt new policies and instruments for fighting towards common goals.

Our narrative review provides an overview of enacted measures to counter the COVID-19 pandemic. The strengths of our work are that it is the first of its kind in literature, gathering several references to documents on the topic and providing a set of epidemiological tools to have a quantitative perspective of the presented results. Nonetheless, it should be noted that it does not include every single possible enacted measure, also lacking a more detailed epidemiological assessment, and the indicators we used are relatively simple when compared with the ones possible in the field of epidemiology, for sake of simplicity and because this manuscript was not meant to be addressed to specialists, but rather to researchers and healthcare professionals intending to learn more on the topic and policymakers willing to have a perspective on the enacted measures.

It should be noted that the goal of this review was to facilitate an understanding of the approach that was held in this context for policymakers to identify behaviors that should be replicated or avoided rather than provide a complete list of measures put in place. For this purpose, our work is the first of its kind, providing a general perspective that can serve as a tool to study Italy’s approach to the COVID-19 pandemic and a starting point for future studies addressing the issue in more detail. Still, it lacks deeper analysis and elaborations of the policies because these are beyond its scope. 

## 5. Conclusions

Starting with an isolated case in northern Italy, the pandemic has severely affected the Italian population. The responses have often been late, incomplete, or ineffective, mainly because Italy was the first country to face such a challenge. 

However, more pandemics will likely happen in the future, and it will be essential to be prepared. Learning from past epidemics, vaccination programs, and mistakes can help define future strategies and build preparedness models adapted to the country’s specificities.

In this paper, we focused only on the salient aspects of the past two years, offering an overview through the narrative review tool and the data’s re-interpretation. We hope that the content of this paper will help build a shared strategy, achieve population engagement, and have an enhanced surveillance system coupled with predefined yet flexible plans to defeat COVID-19 and be ready for future challenges, preserving and protecting public health.

## Figures and Tables

**Figure 1 ijerph-19-15443-f001:**
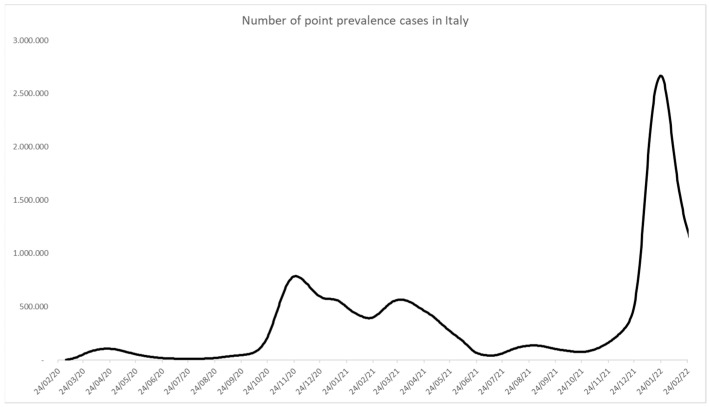
The number of cases in Italy and epidemic waves. This figure was created by elaborating on raw data from ISS [10].

**Figure 2 ijerph-19-15443-f002:**
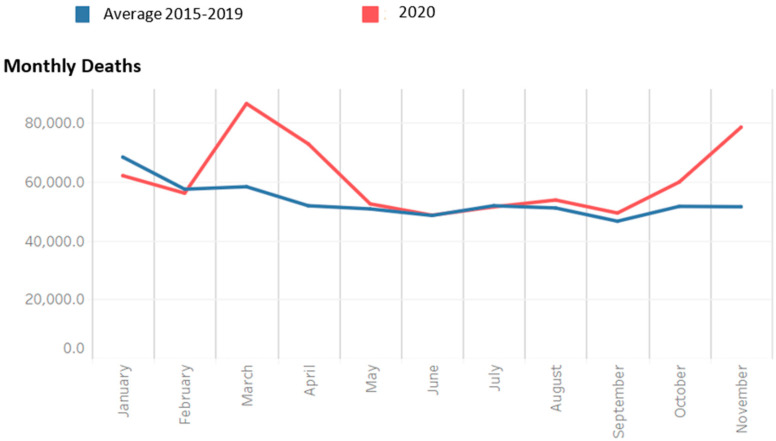
Mortality data for 2020 and average 2015–2019 (ISTAT). This figure was created by elaborating raw data from ISTAT [13].

## Data Availability

Data is available upon request to the corresponding author.

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
