# Peer review of "Narrative Review of the COVID-19 Pandemic’s First Two Years in Italy"

_ijerph, 2022, doi:10.3390/ijerph192315443_

Round 1

Reviewer 2 Report

Dear Authors,

This manuscript presents a very interesting topic. However, some information is needed to improve this manuscript. Minor revisions are needed to improve it. Please see the comments below. Thank you.

General concept comments

1.     Abstract. Abstract already contains the problems, objectives and methods used but has not shown the results of the research. The author is advised to include the results of the study.

2.     Introduction. On line 67 it says "Although several articles were published on pandemic trends and COVID-related issues" can you explain in detail which articles are related to your research? I suggest that the articles can be cited in the introduction.

3.     Materials and Methods.

On line 86-87 "The full texts of the articles considered eligible were evaluated. In addition, the bibliography of included articles was screened to retrieve additional relevant publications"

Would you like to explain the number of articles you got from PubMed?

Would you like to explain the analysis tools do you use in bibliography?

Would you like to explain the analysis tools do you use in narrative review?

Would you like to explain your research framework?

Round 2

Reviewer 1 Report

Dear authors, thanks again for involving me in your work. Please see attached file for my comments. Kind regards
